# Optimization of Extraction Solvent and Fast Blue BB Assay for Comparative Analysis of Antioxidant Phenolics from *Cucumis melo* L.

**DOI:** 10.3390/plants10071379

**Published:** 2021-07-06

**Authors:** Varsha Ravindranath, Jashbir Singh, Guddarangavvanahally K. Jayaprakasha, Bhimanagouda S. Patil

**Affiliations:** 1Vegetable & Fruit Improvement Center, Department of Horticultural Sciences, Texas A&M University, 1500 Research Parkway, Suite A120, College Station, TX 77845-2119, USA; varsharavindranath@tamu.edu (V.R.); singh2014@tamu.edu (J.S.); gkjp@tamu.edu (G.K.J.); 2Department of Food Science and Technology, Texas A&M University, College Station, TX 77845-2119, USA

**Keywords:** *Cucumis melo*, extraction efficiency, Fast Blue, antioxidant activity, phenolics

## Abstract

Melon (*Cucumis melo* L.) fruits contain multiple health-promoting compounds, including phenolic compounds, which are antioxidants. Accurate measurement of antioxidant activities and total phenolic contents (TPCs) require an efficient solvent extraction. In this study, we evaluated free radical scavenging activity and TPC of melon extracts extracted with 22 different solvent combinations. The DPPH scavenging activities were high in 100% methanolic (39.48 ± 0.36 µg g^−1^) and 80% methanolic extracts (38.99 ± 0.44 µg g^−1^). Similarly, the ABTS scavenging activities were high in 100% methanolic (315.11 ± 10.38 µg g^−1^) and 80% methanol extracts (297.39 ± 14.98 µg g^−1^). The Folin–Ciocalteu (F–C) assay is typically used to measure TPC but may be affected by interference from sugars and other compounds. Therefore, we optimized an assay for TPC using Fast Blue (FB) salt and developed a standard operating procedure for microplate analysis using FB. Our analysis of standard samples and comparisons with the F–C assay suggested that the optimized FB assay could be used to measure TPC in fruit and juice samples. Moreover, we successfully detected six phenolic compounds in methanol extracts of melon by LC-HR-QTOF/MS.

## 1. Introduction

Melon (*Cucumis melo* L.) is a commercially important crop that is cultivated in temperate regions of the world, including China, Turkey, and other countries in the Middle East, India, the USA, and Central America [1]. *Cucumis melo* var. *reticulatus* belongs to the Cucurbitaceae family and its fruits are consumed due to their taste, nutritional, and organoleptic attributes [2,3,4,5]. Melons are valuable sources of minerals including potassium and antioxidant compounds such as vitamin C (ascorbic acid), β-carotene, and polyphenols [6]. These compounds are associated with a lower risk of chronic diseases, likely due to their anti-inflammatory [7], antihypertensive [8], antioxidant, anticancer, and antimicrobial properties [9]. Among the phytochemicals found in plants, phenolic acids, and flavonoids have strong free radical scavenging properties, which likely contribute to their health-promoting properties due to their redox properties [10]. Indeed, the consumption of fruits rich in phenolic compounds can limit oxidative stress and reduce the risk of degenerative diseases such as cancer, cardiovascular diseases, and diabetes [11,12,13].

The extraction of phenolic compounds from fruit samples depends on the physicochemical properties of the phenolics and the type of solvent used [14,15]. Indeed, a previous study reported that the profiles of phenolic compounds extracted depend on the polarity of the solvents used for extraction [16]. Phenolic compounds vary in the type, number, and position of functional groups; this results in the variation of chemical properties and thus influences their solubility in different solvents. Hence, selecting the best solvent is a key factor that affects the quality and quantity of extracted phenolic compounds [11,12,17]. 

Solid-to-liquid extraction is commonly used to recover natural antioxidants from plant materials using 100% methanol, 80% methanol, or 100% ethanol [9,18]. For example, an analysis of phenolic compounds and antioxidant activities in melon fruits was carried out using methanol [9] and another study used 80% methanol used to determine the free and bound phenolics in melon fruit pulp [18]. Dry melon peel extracts obtained from aqueous methanol showed high contents of total phenolics compounds compared to the aqueous ethanol extract [19]. A study evaluating the extraction efficiency of solvents was previously carried out using five solvent combinations (acetone, *n*-butanol, 80% ethanol, methanol, and deionized water) and showed that the 80% ethanol extract had the highest total phenolic content (TPC) and antioxidant activity in bitter melon fruits [20]. Another study on cucurbit fruits showed that the most suitable solvent for the recovery of polyphenolic compounds is an aqueous mixture containing acetone, ethanol, methanol, and ethyl acetate [21]. It can be inferred from the literature that the efficiency of extraction is affected by the extraction method, the sample matrix, and the acidity and polarity of the solvent [22].

Antioxidant scavenging activity measured with DPPH and ABTS assays gives a wide range of values in different cultivars when comparing the type of production system or the time frame [23]. The ABTS radical cation is scavenged by hydrophilic and hydrophobic antioxidants, whereas DPPH is scavenged by most hydrophobic compounds in the presence of organic solvents [24]. Therefore, these two assays test different aspects of the antioxidant capacity of samples. Factors such as cultivar, production time, and the type of production system influence the antioxidant levels of melons [25]. The health benefits of melon consumption are attributed to the presence of vitamins, minerals, phenolic metabolites, flavonoids, and alkaloids [26]. However, the recovery of melon phenolic compounds by different solvent combinations have not been explored.

The traditional method to measure TPC in natural products is the Folin–Ciocalteu assay (F–C assay), but the F–C reagents interact and interfere with certain non-phenolic substances such as sugars, aromatic amines, sulfur dioxide, ascorbic acid, or organic acids, which can drastically affect the results. To counter this effect, a single solid-phase extraction cleanup procedure is usually used before the F–C assay. However, this cleanup is costly and time-consuming [27,28]. Another alternative assay to measure TPC in samples is the FB assay. The FB assay is based on the direct reaction of Fast Blue diazonium salts with the active hydroxyl groups present in the phenolic compounds. In the FB assay reaction, the diazonium complex formation occurs in the presence of a base and the reaction is dependent on the position of active -OH groups [29]. This method used to analyze total phenolics seems to be unaffected by the presence of sugars, organic acids, and ascorbic acid present in samples, especially in fruits and beverages that are high in vitamin C. Although the FB assay is useful to evaluate TPC, a protocol for microplate analysis, which uses a low quantity of reagents remains unexplored. 

Herein, we report a systematic investigation of the effects of different extraction solvents on TPC and antioxidant activities. In addition, we developed an optimized standard operating procedure for microplate FB assays and validated the method on 22 solvent extracts from different melon varieties and commercial juices. The present study also involved the identification of phenolic compounds using liquid chromatography coupled with high-resolution quadrupole time-of-flight mass spectrometry (LC-HR-QTOF-MS). 

## 2. Results

### 2.1. Effect of the Extraction Solvent on Free Radical Scavenging Activities

#### 2.1.1. DPPH Scavenging Activity

The results of the DPPH scavenging activity of melon samples extracted with different solvents are shown in Figure 1A and are expressed as micrograms of ascorbic acid equivalents per gram of tissue. The results showed that solvent S8 (100% methanol) had the highest activity (39.48 ± 0.36 µg g^−1^) followed by S7 (methanol:water, 80:20) (38.99 ± 0.44 µg g^−1^). The acidified methanolic extract S3 (methanol: water: formic acid, 50:48:2) showed (33.46 ± 0.92 µg g^−1^) the highest DPPH activity among the methanol:water:acid combinations.

Among the ethanol combinations, S15 (100% ethanol) demonstrated the highest scavenging activity (37.37 ± 1.78 µg g^−1^) followed by S11 (ethanol:water:formic acid, 80:15:5; 29.01 ± 0.30 µg g^−1^) and S14 (ethanol:water, 80:20; 29.27 ± 1.50 µg g^−1^). Samples extracted with 100% acetone (S-22) showed the least DPPH scavenging activity. S16 (acetone:water:formic acid, 50:45:5) demonstrated higher activity (24.86 ± 1.29 µg g^−1^) than the other acetone combinations. The water extract (4.29 ± 1.14 µg g^−1^) showed the lowest scavenging activity and was significantly different from that of the others (*p* < 0.05).

#### 2.1.2. ABTS Scavenging Activity

Results of the ABTS assay are shown in Figure 1B and are expressed as micrograms of ascorbic acid equivalents per gram of tissue. The results demonstrated that extracts of 100% methanol (S8) had the highest activity (315.11 ± 10.48 µg g^−1^), followed by S7 (methanol:water, 80:20; 297.39 ± 14.98 µg g^−1^). Among the ethanol combinations, samples extracted using S14 (ethanol: water, 80:20) showed the highest activity (276.61 ± 13.62 µg g^−1^) followed by 100% ethanol (S15; 254.58 ± 8.83 µg g^−1^). Acetone:water (80:20) combination (S21; 275.83 ± 18.81 µg g^−1^) showed higher activity compared to other acetone combinations.

### 2.2. Optimization of the FB Assay

#### 2.2.1. Optimization of the Base Type, Concentration, and Incubation Time

Four base types viz, NaOH, KOH, Na_2_CO_3_, and K_2_CO_3_, at different concentrations (0.5, 1, and 2 N) were evaluated. Pure gallic acid at 250 µg mL^−1^ conc. was used as a positive control (Table 1). The results demonstrated that among the four bases, KOH at 0.5 N and 1 N showed reactions after 60 min of incubation. In the case of KOH (1 N), we observed a consistent regression coefficient (R^2^ = 0.99) while a reaction time of 120 min showed a linear response in all four replications. Interestingly, higher base concentrations did not favor the reactions and the assay reactions did not precede until 60 min. All graphs pertaining to base optimization are available in Appendix A.

#### 2.2.2. Evaluation of Standards Using the FB Assay

For this analysis, the optimized FB assay was applied to evaluate six standards and the sensitivity of the assay with phenolic compounds is demonstrated (Figure 2).

### 2.3. Comparison of Total Phenolic Contents Using Folin–Ciocalteu (F–C) and FB Assays

The TPC was measured by carrying out F–C and FB assays, gallic acid was used as a standard reference for both the assays (Figure 3). Therefore, the results are expressed as micrograms of gallic acid equivalents (GAE) per gram of tissue. The F–C assay showed higher values when compared with the FB assay, except for water extracts. Results of the FB assay demonstrated that extracts obtained using S1 (water; 94.82 ± 18.39 µg g^−1^) showed the highest TPC followed by extracts obtained using S17 (acetone:water:formic acid (50:48:2); 72.40 ± 6.71 µg g^−1^). The recovery of phenolic compounds in the melon extracts showed that the highest TPC measured using the FB assay was in S1 (water) and the lowest in S22 (acetone) among all the extracts (*p* < 0.05). Interestingly F–C assay results demonstrated that the extracts obtained using S15 (ethanol; 139.49 ± 4.77 µg g^−1^) and S8 (methanol; 137.99 ± 8.02 µg g^−1^) showed higher TPC than other extracts, while S1 (water) showed the least TPC.

### 2.4. Application of the Optimized FB Assay on Melon Varieties

Our results demonstrate that the optimized assay can be applied to accurately measure TPC in melon varieties (Figure 4). Results show the Saurio variety with higher TPC measured by the F–C assay (206.75 ± 8.79 µg g^−1^ gallic acid equivalents (GAE)) compared to the FB assay (196.88 ± 9.00 µg GAE g^−1^), followed by the Tarasco variety, which also showed higher TPC measured by the F–C assay (193.16 ± 4.21 µg GAE g^−1^) than the FB assay (76.45 ± 4.06 µg GAE g^−1^).

### 2.5. Application of F–C and Optimized FB Assays on Commercial Juice Samples

Analysis of juices by the F–C and optimized FB assays showed that the TPC measured by the FB method was higher compared to the values obtained from the F–C method (Table 2). Results of the FB assay showed pomegranate juices J15 (4564.08 ± 114.9 µg GAE g^−1^) and J16 (5634.19 ± 26.83 µg GAE g^−1^) had higher TPC compared to the TPC content measured by the F–C method; J15 (1649.39 ± 40.88 µg GAE g^−1^) and J16 (1756.86 ± 23.23 µg GAE g^−1^).

### 2.6. Identification of Phenolic Compounds by LC/ESI-HR-QTOF-MS

Six phenolics derivatives were identified and their total ion chromatogram, extracted ion chromatogram, and tandem mass spectra are presented in Figure 5A,B. A peak eluted at a retention time (RT) of 1.7 min with *m*/*z* 579.1906 [M + H]^+^ is identified as apigenin-7-*O*-rutinoside based on the mass spectra and by the literature [7]. Another peak that eluted at a RT of 3.7 min was identified as gentisic acid-hexoside isomer 2 (*m*/*z* 317.1207 [M + H]^+^) as previously reported in melon extracts [30]. Similarly, a peak eluted at 5.5 min at RT representing the molecular ion peak at *m*/*z* 611.1606 [M + H]^+^ was identified as rutin. Two peaks that eluted at a RT of 5.5 and 5.9 min were identified as isorhamnetin-3-*O*-glucoside and naringin, having a molecular ion peak at *m*/*z* 479.1184 [M + H]^+^ and *m*/*z* 581.1864 [M + H]^+^, respectively. A peak at RT 7.4 min displayed a molecular ion peak at *m*/*z* 273.0757 [M + H]^+^ and was identified as naringenin and previously reported by [7].

## 3. Discussion

Optimization of extraction of antioxidants from the melon flesh was performed using 22 solvent combinations to understand the impact of extraction solvents on the antioxidant activity. Both DPPH and ABTS scavenging activities were carried out to investigate the effect of solvents on antioxidants extraction. The DPPH is scavenged by the majority of hydrophobic compounds in the presence of organic solvents and the scavenging is observed by the discoloration from purple to light yellow [24]. In our study, sample extracts obtained by using methanol and solvent combinations with water and acid showed higher DPPH activity compared to other solvent combinations. Sample extracts obtained by using ethanol showed lower scavenging activity than methanol combinations. Overall, results of antioxidant activity in the 22 solvents demonstrated 100% methanol extracts to have the highest activity, followed by 100% ethanol extracts and then 100% acetone extracts. The ABTS assay involves the generation of ABTS chromophores by oxidation of ABTS with potassium persulfate; antioxidant compounds prevent the generation of these chromophores. Our results showed that scavenging activities recorded higher when evaluated using the ABTS assay than with the DPPH assay (Figure 1A,B), this may be due to the type of reaction mechanisms—hydrogen atom transfer and single electron transfer, respectively [31]. Moreover, factors like stereo selectivity of radicals or solubility of extracts in different systems affect the extracts capacity to react and quench the different radicals [32,33]. Results of ABTS assays demonstrated that the extracts obtained from solvents combined with water in the ratio of 80:20 (S7, S14, and S21) showed greater antioxidant activity. This may be due to variation in polarity among the different solvent combinations.

The F–C assay has been used as a measure of TPC in natural products, and the basic mechanism is an oxidation–reduction reaction [34]. However, the reaction is slow at acidic pH, and it lacks specificity [35]. A possible alternative is the FB assay that uses the Fast Blue BB diazonium salt (FBBB) in which the diazonium group specifically couples with reactive phenolic hydroxyl (–OH) groups only in the presence of alkali to form stable azo complexes, which can be measured at 420 nm [36]. The azo-based assay (FB method) has higher GAE values than F–C for TPC. A previous study reported a lower value of TPC using the F–C assay compared to the FB assay, for example, for the samples with the addition of ascorbic acid and high-fructose corn syrup [36,37]. In this study, the FB assay was improved by scaling down the amounts of reagents to be performed using the microplate, we optimized various factors to improve the sensitivity of the method and the use of lesser amounts of reagents. For optimizing the base, experiments were carried out using four bases in three varying normalities. A previously published study compared 5% NaOH and 20% Na_2_CO_3_, and showed that 5% NaOH had a faster completion of reaction compared to 20% Na_2_CO_3_ [37]. The use of lower amounts of chemicals to prepare the base solution was preferred. Therefore, in this study we optimized a protocol to determine phenolic compounds using Fast Blue salt. The miniaturized protocol was optimized to be carried out of a 96 well microplate. For the optimized FB assay, 20 µL of 1 N KOH was selected as the base for the 120-min incubation period.

It can be inferred from the results obtained from the application of the optimized Fast Blue assay to evaluate six different phenolics standards that the chemical structure especially the stereochemistry of the -OH group plays an important role in the coupling reaction with FB salt. Previously, the proposed interactions with chlorogenic acid, caffeic acid, and flavonoids in the NaOH buffer showed the formation of the azo complex in active ortho and para positions [37]. From the proposed mechanisms, it can be understood that the higher number of available reactive -OH groups in the compound that are present in ortho and para positions may influence the reaction times (incubation). The quercetin standard had a linear reaction at 30 min, followed by chlorogenic acid, naringenin, and catechin hydrate at 60 min, and lastly gallic acid at 90 min while caffeic acid showed poor linearity. However, in the presence of KOH (1 N), quercetin showed the highest linearity at 20 min (R^2^ = 0.99) due to the presence of more reactive -OH groups at the ortho and para positions.

Previous studies on the Fast Blue assay were carried out to assess beverage and juice samples, using the FB method the assay mixtures in borosilicate tubes were prepared and the mixture was transferred to a microplate; this optimized assay does not require preparation in separate tubes [27,36,37,38,39]. The alternative method used to measure total phenolic content is the Folin–Ciocalteu assay. The F–C assay measures all compounds readily oxidizable under the reaction conditions inclusive of monophenols and certain substances that are non-phenols or proteins also tend to react under these conditions [40]. The high TPC results obtained from the F–C assay indicated the presence of non-phenolic compounds. Extracts obtained from S1 (water) also showed low TPC; this indicates that water extracts may contain several impurities that affect the reaction and detection of phenolic compounds in the extract. Among all combinations used, methanol (100%) and ethanol (100%) combinations showed the highest phenolic compounds recovery. Melon extracts analyzed with different solvents showed highest TPC in the order water > methanol > methanol + water > ethanol +water + acid > acetone + water + acid > acetone+ water > acetone (Figure 3). Previous studies also showed higher TPC in water extracts [41,42]. However, higher extraction yield does not always reflect higher antioxidant activity, because the antioxidant activity also depends on the active antioxidant compounds present in the extract [42]. This agrees with our results, as water showed the highest TPC when measured using the FB assay but the lowest antioxidant activity when measured using DPPH and ABTS antioxidant assays.

From the evaluation of 22 solvent combinations used to extract melon phenolics, it was demonstrated that water extracts were suitable to measure TPC using the optimized FB assay efficiently than other combinations, and, therefore, water was used as a solvent for the extraction of melon samples. To evaluate the TPC content in six melon varieties (Syngenta) harvested from Uvalde, TX were examined using the optimized FB assay and compared with the F–C assay. The result showed the F–C assay had higher TPC than the FB assay (Figure 4). Some melon varieties (Syngenta) showed comparatively lower TPC from the FB assay, which may be influenced by other compounds that interfere with the detection of phenolic compounds (such as organic acids, sugars, and ascorbic acid). An analysis of beverages and grains obtained similar results, showing higher F–C value results compared to the FB value [31]. To further study, applications of the optimized assay we examined commercial juices and compared the total phenolic content measured by F–C and optimized FB methods. Our study shows that pomegranate juices measured high TPC among all juices analyzed by FB and F–C assays. The results obtained suggested that there may be a high concentration of non-phenolic compounds in the sample extracts obtained from commercial juices. A similar study comparing the FB and F–C assays used fresh pomegranate extracts and showed higher TPC measured by the FB method (193 mg GAE 100 g^−1^) compared to the F–C method (161 mg 100 g^−1^) [37]. This suggested that non-phenolic compounds and reducing sugars GAE naturally present or added to juice mixes contribute to higher F–C values. The same study also showed that the results from the FB method were higher than the F–C method, which agrees with our results. Another notable observation is that the quantity of the sample needs to be modified according to the presence of phenolic compounds. For example, melon extracts used for FB assay were expected to contain low levels of phenolic compounds therefore higher sample quantity (40 µL) was used for the analysis of phenolic compounds while commercial juices such as pomegranate and prune juice was expected to be rich in phenolic compounds and therefore a lower sample quantity (10 µL) of juice sample was used for the FB assay.

It has been suggested that the chemical composition of melon and other fruits is strongly influenced by the physiological state of the plants and by the environmental parameters and by the genotype [30]. Previously, a study investigated the anti-inflammatory activity of phenolic compounds in melon peel and pulp and represented a metabolic profile of the various compounds detected in ethanol extracts using UPLC-DAD-MS/MS in the negative ionization mode [7]. In our study, the identification of compounds in melon extracts was carried out using LC-HR-ESI-QTOF-MS and six compounds were successfully identified.

## 4. Materials and Methods

### 4.1. Chemicals

ACS and LCMS grade solvents were used for extraction and liquid chromatography (LC) analysis. Methanol, acetonitrile, acetone, ethanol, formic acid, phosphoric acid, sodium carbonate, potassium hydroxide 2,2-diphenyl-1-(2,4,6-trinitrophenyl) hydrazyl (DPPH), Folin–Ciocalteu (F–C) reagent, 2,2’-azinobis (3-etylbenzothiszoline-6-sulphonic acid) diammonium salt (ABTS), FB B salt, quercetin, gallic acid, naringin, naringenin, and chlorogenic acid were obtained from Sigma (St. Louis, MO, USA). Bases potassium hydroxide (KOH), sodium hydroxide (NaOH), sodium carbonate (Na_2_CO_3_), and potassium carbonate (K_2_CO_3_) were obtained from Fisher Scientific (Pittsburg, PA, USA).

### 4.2. Plant Material and Extraction Solvents

Melons used for phenolics extraction were obtained from the H-E-B supermarket (College Station, TX, USA). For this experiment, two melon fruits were used, and duplicate samples were used for each solvent combination (*n* × 2). Each melon was washed and cleaned under running water for 1 min, dried, cut into two halves, deseeded, peeled, chopped into small cubes, and then blended (Oster 6684 12-Speed Blender) to obtain melon juice. Freshly blended melon sample (10 g) was measured into clean 50 mL centrifuge tubes (VWR, 50 mL tubes) and mixed with 10 mL of solvent. We used 22 solvent combinations and designated each solvent combination with the letter S and a number: S1, water; S2, methanol:water:formic acid (50:45:5); S3, methanol:water:formic acid (50:48:2); S4, methanol:water:formic acid (80:15:5); S5, methanol:water:formic acid (80:18:2); S6, methanol:water (50:50); S7, methanol:water (80:20); S8, 100% methanol; S9, ethanol:water:formic acid (50:45:5); S10, ethanol:water:formic acid (50:48:2); S11, ethanol:water:formic acid (80:15:5); S12, ethanol:water:formic acid (80:18:2); S13, ethanol:water (50:50); S14, ethanol:water (80:20); S15, 100% ethanol; S16, acetone:water:formic acid (50:45:5); S17, acetone:water:formic acid (50:48:2); S18, acetone:water:formic acid (80:15:5); S19, acetone:water:formic acid (80:18:2); S20, acetone:water (50:50); S21, acetone:water (80:20); S22, 100% acetone. Sample tubes were vortexed for 30 s, homogenized for 1 min at 9000 rpm, and sonicated for 60 min (Cole-Parmer Ultrasonic cleaner 8893). Sample tubes were then centrifuged (VINTAGE Beckman J2-21 208V 30A 341735 Refrigerated Centrifuge) at 12,000 rpm. The supernatant was then filtered using Whatman filter paper Grade 1 (32 mm). In order to ensure complete extraction, the residue obtained from the 1st extraction was re-extracted again using 10 mL of the respective solvents. The extracts (1st and 2nd extractions) were then pooled, collected in clean tubes, and stored at −20 °C. These extracts were further used to measure TPC and radical scavenging activities.

### 4.3. Antioxidant Assays

#### 4.3.1. DPPH Assay

Melon extracts of different varieties were measured for scavenging activity using the 2,2- diphenyl-1-picrylhydrazyl (DPPH) free radical. For this assay, 40 µL aliquots of melon extracts obtained from 22 solvents were pipetted into the sample wells (in triplicate), the total volume of each well was adjusted to 100 µL with 60 µL of methanol. Then, 180 µL of methanolic DPPH solution was added to sample wells and incubated for 30 min in the dark and the absorbance was recorded at 515 nm using a microplate reader (Bio Tek Instruments, Winooski, VT). Standard ascorbic acid (20 µg mL^−1^) was used for the positive control to prepare the calibration curves (increasing volume 10, 20, 30, 40, 50, and 75 µL). Results were expressed in micrograms of ascorbic acid equivalents g^−1^ of the sample. The DPPH free radical scavenging activity was measured according to our published protocol with a slight modification [29].

#### 4.3.2. ABTS Radical Scavenging Activity

The 2,2’-azino-bis-(3-ethylbenzothiazoline-6-sulphonic acid) (ABTS) reagent was prepared by mixing 0.165 g of potassium persulfate and 0.193 g of ABTS in Nanopure water. After overnight incubation in the dark, a green colored solution was obtained. Fresh ascorbic acid standard (50 µg mL^−1^) was prepared and used for the assay as a positive control and to prepare the calibration curve (increasing volume 5, 10, 15, 20, 25, and 30 µL). The ABTS radical cation eliminating activity was determined according to our published protocol [43]. For this assay, 10 µL aliquots of melon extracts were pipetted into the sample wells (in triplicate), followed by 90 µL of methanol to adjust the volume in the sample wells. The reaction was initiated by adding 180 µL of ABTS stock solution into the sample mixture and absorbance was recorded at a wavelength of 734 nm using the microplate reader. Results were expressed as micrograms of ascorbic acid equivalent g^−1^ of the sample.

### 4.4. Folin–Ciocalteu Method to Measure Total Phenolic Contents

The TPC in the different melon extracts was determined by the F–C assay, as previously described, with slight modifications [44]. To determine the total phenolic content microplate in the melon extracts, standard gallic acid (50 µg mL^−1^) was pipetted in the first 3 columns of the 96-well in increasing concentrations (10, 20, 30, 40, 50, 75, and 100 µL), then the volume in standard wells was adjusted accordingly with Nanopure water to obtain a total volume of 180 µL. In the sample wells, 20 µL aliquots of extract were pipetted. The volumes in each sample wells were also adjusted to 180 µL with Nanopure water. Then, 20 µL of Folin–Ciocalteu working solution was added to all wells and then incubated the plates in the dark briefly for 10 min followed by addition of 50 µL of sodium carbonate in all wells and further incubated for 20 min in the dark. The absorbance was monitored at 760 nm using the microplate and TPC was expressed as micrograms of ascorbic acid equivalent g^−1^ of the sample.

### 4.5. Optimization of the Microplate Assay Using Fast Blue B Salt

The optimization parameters included the selection of the base, base concentration, standards, incubation time, volume of sample, and addition of water. The Fast Blue B salt (FBBS) solution (0.01%) was prepared by dissolving 10 mg of FBBS in 100 mL of Nanopure water.

#### 4.5.1. Optimization of Base Type and Concentration

For the microplate analysis, solutions of bases at different normalities were prepared using Nanopure water. Three concentrations (0.5, 1, and 2 N) of NaOH, KOH, Na_2_CO_3_, and K_2_CO_3_ were evaluated. Gallic acid (250 µg mL^−1^) was pipetted in increasing volumes (10, 20, 30 40, 50, 75, and 100 µL aliquots) into the wells of a microplate, the volume was adjusted with 200 µL of Nanopure water in all wells. Then, 20 µL of FB solution (0.01%) was pipetted into all wells followed by 10 min of incubation in a dark environment. Further, varying concentrations of bases were pipetted into different sample wells and allowed to incubate. The resulting color was measured spectrophotometrically at 420 nm. Quadruplicate readings were taken, and absorbance was measured at 0, 30, 60, 90, and 120 min. The regression equations at each time period were compared to understand the effect of the base at each time point.

#### 4.5.2. Evaluation of Standards Using the Optimized Fast Blue Assay

For evaluating the sensitivity of the optimized FB assay six phenolic standards viz; quercetin, chlorogenic acid, caffeic acid, catechin hydrate, naringenin, and gallic acid were prepared at 250 µg mL^−1^ and used to determine the linearity of the standards (increasing volume 10, 20, 30, 40, 50, 75, and 100 µL) at different incubation periods. Absorbance was measured at 0, 30, 60, and 90 min. The regression equation at each time point was compared to understand the sensitivity of standards.

### 4.6. Application of The Optimized FB Assay to Measure Total Phenolic Content in Melons

Fruits from six melon varieties, Tarasco, T-Rex, Accolade, Mamut, Saurio, and Sweet spring was blended and extracted based on the results of extraction solvent efficiency from the initial experiments.

Blended melon juice samples (10 g) were extracted using Nanopure water twice to ensure complete extraction. The supernatant was pooled and filtered using Whatman filter paper Grade 1 (32 mm), the collected extracts were stored at −20 °C for further use. An FB assay was carried out according to the developed protocol. To perform the Fast Blue assay on a clean 96-well microplate, 40 µL aliquots of sample extracts were pipetted, followed by the addition of 160 µL of Nanopure water, then 20 µL of freshly prepared FBBS solution (0.01%) was added in all wells. The plate was covered with aluminum foil and incubated for 10 min in the dark, then 20 µL of 1 N KOH was added to the sample mixture in all wells. The plate was further incubated for 120 min to allow the reaction to occur. After 120 min, absorbance was recorded at 420 nm in a microplate reader. To compare results, the F–C assay was carried out according to a previously published protocol [44] with slight modifications. Melon extracts (20 µL) were pipetted into sample wells followed by Nanopure water (180 µL), then 20 µL of the F–C reagent (prepared by mixing 20 mL of the F–C reagent with 20 mL of Nanopure water). The plates were covered with aluminum foil and incubated for 10 min. Briefly, after 10 min of incubation, 50 µL of saturated sodium carbonate solution (prepared by dissolving 140 mg of sodium carbonate in 1 L of Nanopure water) was added to all the wells. The plate was further incubated for 20 min after which absorbance of the samples were recorded at 760 nm.

### 4.7. Application of The Optimized FB Assay to Assess Total Phenolic Content in Commercial Juices

The application of the optimized assay was checked by evaluating TPC in commercial juice samples to confirm the reproducibility of the assay. Pure juices of different brands available commercially in the H-E-B supermarket (College Station, TX, USA). Juices were purchased and extracted according to previously published paper [45] with slight modifications, for this analysis 100% methanol (methanol: juice in the ratio 2:1) was mixed with juices for extraction. Juice samples and reagent concentration were adjusted according to the activity of the sample. For example, the juice obtained from white grape juices (J1 and J2) had lower phenolic content compared to the pomegranate juices (J15 and J16). Therefore, for the FB assay of J1-J12, 40 µL of the sample extract followed by adding 160 µL of Nanopure water and 20 µL of FB solution was added. Samples were kept for a 10 min incubation and later 20 µL of 1 N KOH was added to the reaction mixture. For juices J13 to J16, 10 µL of sample extract was used followed by 190 µL of Nanopure water, 20 µL of FB solution, and, after 10 min of incubation, 20 µL of 1 N KOH solution was added. The F–C assay was also performed on juice samples, according to the protocol [44].

### 4.8. Identification of Phenolic Compounds by Liquid Chromatography/Electrospray Ionization High-Resolution Quadrupole Time-of-Flight Tandem Mass Spectrometry (LC/ESI-HR-QTOF-MS)

To identify phenolic compounds, the lyophilized melon sample (0.250 g) was extracted with 5 mL of various solvent combinations. The sample mixture was vortexed (1 min), homogenized (1 min), sonicated for 60 min, and centrifuged (Beckman Model TJ-6) at 4480× *g* for 15 min. The sample extracts were used for the identification of phenolics using Agilent 1290 liquid chromatography (Santa Clara, CA, USA) equipped with a diode array detector and coupled to a maXis Impact high-resolution mass spectrometer (Bruker Daltonics, Billerica, MA, USA). The chromatographic separation was carried out on a Zorbax Eclipse Plus C18 column (100 mm × 2.1 mm, 1.8 μm) at a flow rate of 100 μL min^−1^. The mobile phase was composed of (A) 0.1% formic acid in water and (B) 0.1% formic acid in acetonitrile. The solvent gradient was 80–20% B (0 min), 50–50% B (2 min), 38–62% B (3 min), 32–68% B (7 min), 28–72% B (3 min), 10–90% B (12 min), 80–20% B (14 min), and 80–20% B (16 min).

Mass spectral analyses were performed in the positive ionization mode according to our previous methodology. The MS and bbCID (bond band collision-induced dissociation) spectra were acquired at the *m*/*z* range of 25–2000 amu. For the ion source capillary, the voltage was 4200 V, with the end plate offset at 500 V, and the nebulizer gas pressure was 2.8 bar. Nitrogen was used as a nebulizer and drying gas with the 8.0 L/min flow rate and the temperature was kept at 220 °C. The transfer time of the source was 72 μs and the prepulse storage time was 1 μs. The quadrupole MS and bbCID collision energy were set at 5 eV and 70 eV, respectively. The mass spectrometer calibration was performed by sodium formate solution (1 mM sodium hydroxide and water:isopropanol (1:1) with 0.2% formic acid) at the end of each run using a Cole Palmer syringe pump (Vernon Hills, IL, USA) [46,47].

### 4.9. Statistical Analysis

For this study, two melon fruits were used and pooled together to form one homogenous sample. Each solvent extract from the melon samples was prepared in duplicate and the results were presented as means ± SE. For analysis of commercial juices, 16 commercial juice brands were selected and duplicate samples from each were used for the experiment. Analysis of variance and significant differences among means were tested by a one-way ANOVA and a Student’s *t*-test was used to compare the means using JMP (version 14.0 for Windows SAS Inc., Cary, NC, USA). The regression analysis (R^2^ values) was conducted using MS Excel (Microsoft Office 365).

## 5. Conclusions

In the present study, the optimized FB assay proved to be repeatable and showed negligible differences between each replication. The protocol developed in this study can be successfully applied to measure TPC in different melon extracts and commercial fruit juices. Moreover, the optimized FB assay was miniaturized to examine samples efficiently in a short time. One of the limitations noticed in the FB assay is that reaction time varies with the type of phenolic compounds present in the sample extract. The antioxidant activity was found to be influenced by the polarity and type of phenolic compounds present in the extracts. Six phenolic compounds were identified based on chromatographic separation and MS/MS fragmentation using LC–MS. These phenolic compounds may be major contributors to the antioxidant activity of the melon as the results showed 100% methanol extracts had high free radical scavenging activity. This work provided an insight into the efficiency of solvents to obtain phenolic compound rich extract from the melon and for the optimized assay to measure the rapid and accurate antioxidant potential of melon fruits.

## Figures and Tables

**Figure 1 plants-10-01379-f001:**
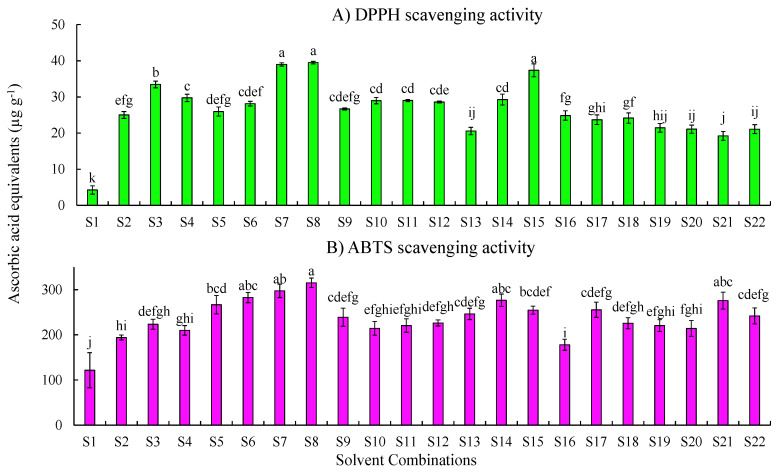
Free radical scavenging activity of cantaloupe extracts measured by the (**A**) DPPH assay and (**B**) ABTS assay. Solvent combinations used: S1—water, S2—methanol:water:formic acid (50:45:5), S3—methanol:water:formic acid (50:48:2), S4—methanol:water:formic acid (80:15:5), S5—methanol:water:formic acid (80:18:2), S6—methanol:water(50:50), S7—methanol:water (80:20), S8—methanol (100%), S9—ethanol:water: formic acid (50:45:5), S10—ethanol:water:formic acid (50:48:2), S11—ethanol:water:formic acid (80:15:5), S12—ethanol:water:formic acid (80:18:2), S13—ethanol:water (50:50), S14—ethanol: water (80:20), S15—ethanol 100%, S16—acetone:water:formic acid (50:45:5), S17—acetone:water:formic acid (50:48:2), S18—acetone:water:formic acid (80:15:5), S19—acetone:water:formic acid (80:18:2), S20—acetone:water (50:50), S21—acetone: water (80:20), and S22—acetone (100%). Results are represented as mean ± SE (µg g^−1^) ascorbic acid equivalents. Means with the same letters indicate no significant differences (*p* ≤ 0.05) among the solvent combinations.

**Figure 2 plants-10-01379-f002:**
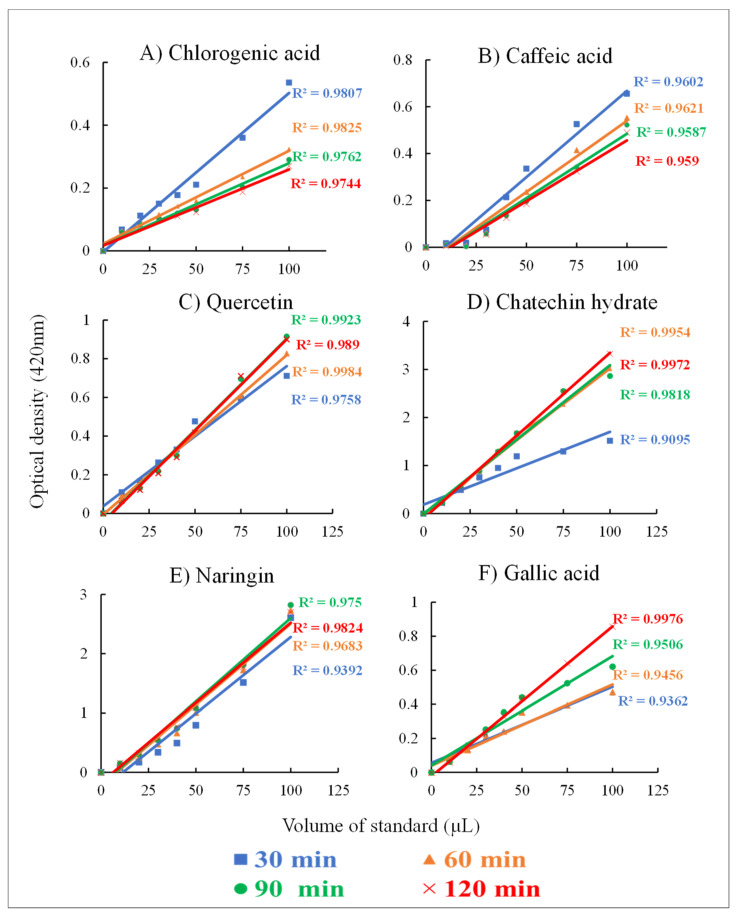
Reaction rate and incubation times evaluated using the optimized FB assay on standards. Linear regression lines were plotted to obtain R^2^ at four-time points. Phenolic standards: (**A**) chlorogenic acid, (**B**) caffeic acid, (**C**) quercetin, (**D**) catechin hydrate, (**E**) naringenin, and (**F**) gallic acid. Absorbance was read at 420 nm with kinetic readings. All standards concentrations were 250 µg mL^−1^ and alkali medium KOH (1 N) and 0.01% FB assay solution was used for the assay.

**Figure 3 plants-10-01379-f003:**
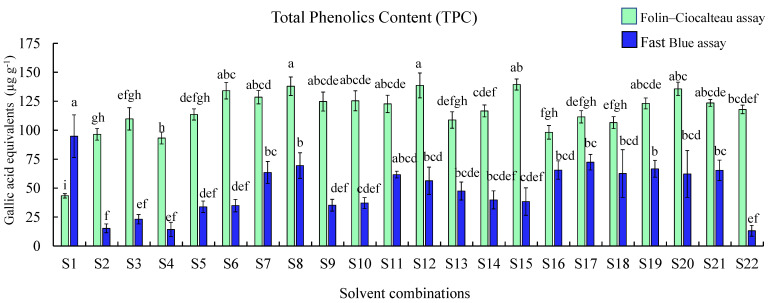
Total phenolic content measure by the Folin–Ciocalteu and Fast Blue assay in cantaloupe extracts. Solvent combinations used S1—water, S2—methanol:water:formic acid (50:45:5), S3—methanol:water:formic acid (50:48:2), S4—methanol:water:formic acid (80:15:5), S5—methanol:water:formic acid (80:18:2), S6—methanol:water (50:50),S7—methanol:water (80:20),S8—methanol 100%, S9—ethanol:water: formic acid (50:45:5), S10—ethanol:water:formic acid (50:48:2), S11—ethanol:water:formic acid (80:15:5), S12—ethanol:water:formic acid (80:18:2), S13—ethanol:water (50:50), S14—ethanol: water (80:20), S15—ethanol 100%, S16—acetone:water:formic acid (50:45:5), S17—acetone:water:formic acid (50:48:2), S18—acetone:water:formic acid (80:15:5), S19—acetone:water:formic acid (80:18:2), S20—acetone:water (50:50), S21—acetone:water (80:20), and S22—acetone 100%. Results are represented as mean ± SE (µg g^−1^) gallic acid equivalents (GAE) for each solvent, different letters indicate significant differences (*p* ≤ 0.05) among the solvent combinations.

**Figure 4 plants-10-01379-f004:**
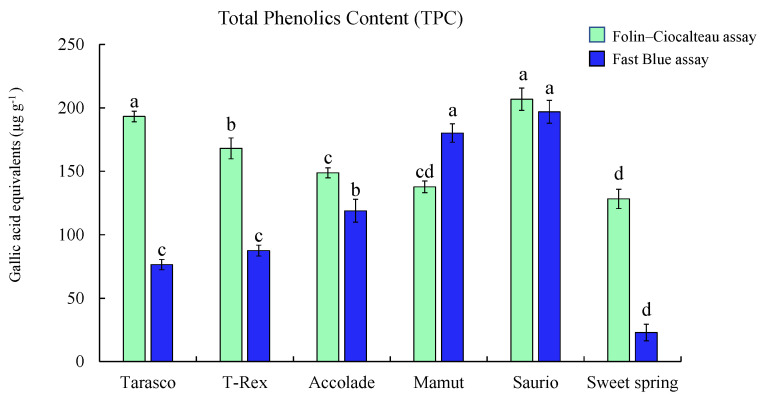
Application of the optimized assay on melon samples (Syngenta) obtained from Uvalde, Texas. Total phenolic content measured using the FB assay and Folin–Ciocalteu assay. Results are represented as mean ± SE (µg g^−1^) gallic acid equivalents for each variety, different letters indicate significant differences (*p* ≤ 0.05) among the melon varieties.

**Figure 5 plants-10-01379-f005:**
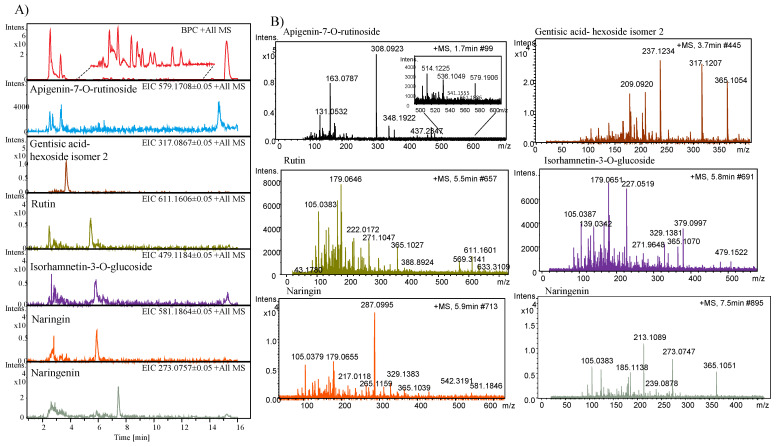
(**A**) Total ion chromatogram and extracted ion chromatograms by LC-HR-ESI-QTOF-MS and (**B**) tandem mass spectra of phenolic compounds identified from the methanol extracts of melon.

**Table 1 plants-10-01379-t001:** Optimization of base concentrations for the FB assay carried out at different incubation times. The average R^2^ results represented below are calculated by the values obtained from the simple linear regression coefficients (R^2^) of four replications. Absorbance of the plates were recorded at 420 nm.

Base	Concentration (N)	0 min	30 min	60 min	90 min	120 min
**NaOH**	0.5	0.832	0.956	0.970	0.981	0.990
	1	0.858	0.960	0.985	0.991	0.995
	2	0.789	0.906	0.902	0.945	0.969
**KOH**	0.5	0.822	0.975	0.991	0.989	0.991
	1	0.796	0.920	0.943	0.965	0.992
	2	0.804	0.808	0.923	0.907	0.901
**Na_2_CO_3_**	0.5	0.898	0.955	0.989	0.991	0.990
	1	0.858	0.967	0.989	0.991	0.995
	2	0.831	0.980	0.988	0.978	0.973
**K_2_CO_3_**	0.5	0.827	0.854	0.937	0.970	0.982
	1	0.662	0.835	0.931	0.959	0.976
	2	0.735	0.876	0.943	0.967	0.977

**Table 2 plants-10-01379-t002:** Total phenolic content of commercial juice extracts measured by the optimized FB and Folin–Ciocalteu (F–C) assays. Results represented as mg g^−1^ ± SE as gallic acid equivalents (GAE). For each juice, means with the same letters indicate no significant differences (*p* ≤ 0.05) in the different juices.

Juice ID	FB Assay (µg GAE g^−1^)	F–C Assay (µg GAE g^−1^)
J1	301.18 ± 8.33 ^f,g^	347.01 ± 9.24 ^h,i^
J2	41.42 ± 1.88 ^h^	127.31 ± 12.39 ^k^
J3	93.56 ± 7.49 ^g,h^	150.87 ± 13.88 ^k^
J4	321.7 ± 12.81 ^f^	379.82 ± 18.7 ^g,h^
J5	135.31 ± 9.3 ^f,g,h^	289.17 ± 8.02 ^i,j^
J6	267.77 ± 28.62 ^f,g,h^	298.07 ± 19.13 ^i,j^
J7	676.84 ± 30.29 ^e^	287.5 ± 17.23 ^i,j^
J8	950.11 ± 10.41 ^c,d^	267.9 ± 12.4 ^1,j^
J9	738.28 ± 22.13 ^d,e^	567.89 ± 15.34 ^f^
J10	1010.97 ± 30.29 ^c^	836.12 ± 26.62 ^c^
J11	343.88 ± 18.24 ^f^	409.07 ± 19.07 ^g^
J12	594.3 ± 19.93 ^e^	435.57 ± 11.54 ^g^
J13	716.92 ± 11.44 ^e^	798.65 ± 20.32 ^d^
J14	730.77 ± 13.64 ^e^	659.77 ± 21.1 ^e^
J15	4565.08 ± 114.91 ^b^	1649.39 ± 40.88 ^b^
J16	5634.19 ± 26.83 ^a^	1756.86 ± 23.23 ^a^

## Data Availability

The authors confirm that the data supporting the findings of this study are available within the article.

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
