# Peer review of "Optimization of Extraction Solvent and Fast Blue BB Assay for Comparative Analysis of Antioxidant Phenolics from Cucumis melo L."

_plants, 2021, doi:10.3390/plants10071379_

Round 1

Reviewer 1 Report

The authors presented here an interesting paper describing the improved approaches to identify phenolic compounds and compare their properties such as antioxidant activities.

 The writing needs to be improved, some fond formats need to be corrected and the results are not very convincing me.

Major problems:

  1. I’m not quite sure when you indicated in your text that ‘The results indicate that the optimized assay can be applied to accurately measure TPC in melon varieties’, what results showed that?
  2. In Table 2, many juices had higher TPC when measured by F-C assay, I’m very confused by this table and what do you want to show and prove here. It still unclear to me that which method is better when you compare the two methods.
  3. ‘The high TPC results obtained from the F-C assay indicated the presence of non-phenolic compounds and higher values may be due to the reaction with interfering substances present in the extract.’ Where is the evidence to show this conclusion?

Minor problems:

  1. I don’t quite understand the first sentence, what ‘temperature regions’ do you really mean?
  2. ‘Melons are valuable sources of minerals (including potassium) and antioxidant compounds such as vitamin C (ascorbic acid), β-carotene, and polyphenol, [6]. ‘ Last coma is not necessary here.
  3. ‘Na2CO3 ‘and ‘K2CO3 ‘in your paragraph should be Na2CO3 and K2CO3
  4. R2=0.99 should be R2=0.99
  5. What does different letters represent as regarding to the significant difference, please explain in detail.
  6. Figure 2, the R2 values listed in figures are quite confused, not sure which represents which.

Reviewer 2 Report

The manuscript illustrates how some commonly used assays for the determination of antioxidant power or total phenols are affected by both the extraction and the chemical nature of the antioxidant or phenolic molecules contained in the samples. A lot of work has been carried out and the results are interesting. However, the manuscript in its present form needs deep reorganization and clarification. Especially the Abstract, the Materials and Methods and the Results sections should be rewritten, with particular attention to the following points:

-In the Abstract, the sentences at Lines 22-26 are too generic. It is generally acknowledged that the solvent nature affects the extraction efficiency; moreover, the manuscript does not concern only the issue of the extraction solvent. The Authors should better mention the specific optimization results that they obtained in the present work.

-Units must be clearly indicated. For example, in the manuscript, ‘μg of ascorbic acid equivalents (AAE) per gram of sample’ is reported at Line 423, ‘μg mL-1 AAE of juice sample’ is reported at Line 425, ‘mg kg-1’ is reported in the abstract, and ‘ascorbic acid equivalence mg kg-1’ is reported in the graph of Figure 1. Similar non-uniform units are reported for TPC. The unit ‘mg L-1’ is reported as ‘ppm’ in Line 183. The same notations should be used throughout the manuscript. Please check and clarify. Please also check the unit (µg or µg L-1) at Line 153.

-In the Materials and Methods, it is correct to report the reference for each protocol that can be found in the literature (see for example Line 424). However, complete information must be provided to the readers, and all the details of all the protocols that have been used in the work should be reported also in the text.

-All the graphs concerning the optimization of the FB method as described in subsection 2.2.1 should be reported.

-In subsection 4.9 Statistical analysis, please mention the regression analysis.

-In the Results, some sentences do not describe results and should be better moved to the Discussion section. For example, Lines 167-170; Lines 188-190; Lines 232-236.

-In Table 1, the expression 'average of the regression equations' in the caption is not clear. If the values reported in Table 1 are the mean values of the Pearsons' coefficients, this must be specified. Also, the coefficients should be reported with 3 significant digits.

Lines 78-79. Please rephrase: 'alkaline medium' means 'in the presence of base'. Similarly, at Lines 299-300 ‘in the presence of alkaline’ should be replaced by 'in the presence of alkali' or 'in alkaline medium'.

Lines 140-141; Lines 253-254. Please rephrase.

Line 145. Please specify which type of activity.

Lines 288-295. The values obtained with the ABTS assay are about 10-fold higher than those obtained with the DPPH assay. Can all the factors that have been mentioned in the text account for such a large difference?

Line 465. According to Table 1, 120 minutes incubation time was also included. According to Table 1, four plate readings were taken.

Lines 542-543. This sentence is unclear; please rephrase.

Reviewer 3 Report

- It is well known that the polarity of the solvent affects the extraction process of phenolic compounds.

  • There are a lot of work on this topic  eg. Phytochemical Analysis and Antioxidant Potential of Cucumis Melo Seeds. Int. J. Life. Sci. Scienti. Res., 3(1): 863-867, 2017
  • Phenolic Profile and Antioxidant Activity of Melon (Cucumis Melo L.) Seeds from Pakistan, Foods 2016, 5, 67; doi:10.3390/foods5040067
  • PHYTOCHEMCAL STUDIES OF BIOACTIVE COMPOUNDS FROM Cucumis melo LINN. Hygeia.J.D.Med.9 (2) January 2017; 1-6

- the manuscript does not add new information

- The antioxidant activity is not affected only by flavonoids and phenolic acids, but also other compounds present in plants.

- More information should be given about the conditions of the MS / MS analysis performed

- the discussion of results and conclusions need to be improved.

Round 2

Reviewer 1 Report

I'm satisfied with the authors' answers while still noticed some grammar errors such as 'The FolinCiocalteu (F-C) assay is typically use to measure TPC, but can show interference from sugars and other compounds.', 'Our analysis of standard samples and comparisons with the F-C assay suggested that the optimized FB assay could use to measure TPC in fruit and juice samples.' in abstract and other errors in the following text. The authors needs to check the writing carefully.

Reviewer 2 Report

The manuscript was much improved. Some minor points follow:

Line 117. After “Figure 1 (A)” it would be advisable to add “and are expressed as µg ascorbic acid equivalents per gram of tissue”.

Line 131. Similarly, after “Figure 1B” it would be advisable to add “are expressed as µg ascorbic acid equivalents per gram of tissue”.

Line 187. Similarly, after “Figure 3” it would be advisable to add “Therefore, the results are expressed as µg gallic acid equivalents (GAE) per gram of tissue”.

Please replace “ppm” by “µg mL-1” in Line 157 and wherever “ppm” is reported.

Subsection 2.4 and Table 2. Please replace “µg g-1 GAE” by “µg GAE g-1”.

Figure 5. Chromatograms are unreadable. Figure 5 was much better in the first version of the manuscript.

Reviewer 3 Report

I accept changes in the manuscript. The introduced changes increased the value of the manuscript.
